# External Two Stage DEA Analysis of Bank Efficiency in West Balkan Countries

Nada Milenković [image: ORCID], Boris Radovanov *[image: ORCID], Branimir Kalaš and Aleksandra Marcikić Horvat

Faculty of Economics in Subotica, University of Novi Sad, 24000 Subotica, Serbia;
nada.milenkovic@ef.uns.ac.rs (N.M.); branimir.kalas@ef.uns.ac.rs (B.K.);
aleksandra.marcikic.horvat@ef.uns.ac.rs (A.M.H.)
*  Correspondence: boris.radovanov@ef.uns.ac.rs

**Abstract:** Since the beginning of the application of the Data Envelopment Analysis (DEA) model in various areas of the economy, it has found its wide application in the field of finance, more specifically banks, in the last few years. The focus of this research was to determine the sustainability of the intermediate function of banks, especially in recent years when interest rates on deposits have been at a minimum level. The research was divided into two parts, wherein the first part determined the efficiency of the intermediate function of banks in the countries of the Western Balkans in the period from 2015 to 2019. The second part approached the regression analysis in which we determined the influence of the bank size, type of bank, and mergers and acquisitions (M&A) activity on the defined efficiency. In the first stage we applied the output-oriented DEA model using deposits, labor costs, and capital as input variables; on the other side, we used loans and investments as output variables. We used data from the revised financial statements of the banks operating in Serbia, Bosnia and Herzegovina, Montenegro, North Macedonia, and Albania. The results of our study showed that there is a difference in efficiency levels between countries and within countries in the considered time period. Furthermore, Tobit regression analysis showed a significant and negative influence of the bank type and M&A on relative technical efficiency of banks, and a positive and significant relationship between bank size and relative efficiency. These findings suggest that large commercial banks can sustain on the West Balkan market. It is to be expected that less efficient small banks will be taken over by large and more efficient banks.

**Keywords:** bank efficiency; two stage analysis; DEA; Tobit regression

## 1. Introduction

Financial systems of the Western Balkan countries rely on banking institutions. The main gain of the banking institutions is to generate profit by mediating between entities with excess funds and those who lack of funds. In recent years, the interest rates paid on deposits have been declining to the point of becoming symbolic. These negative interest rates are influencing the banks' operations in several ways [1]. It is more profitable for individuals to invest excess financial savings in more profitable placements rather than deposit them in banks. That could lead to the withdrawal of deposits from banks, which could endanger the banks' liquidity [2]. As a result, banks' deposit potential could be reduced as well as the loan potential which is generated by the deposit potential. In this way, the monetary policy unintendedly affects the generation of loans and, indirectly, the banks' profit [3,4]. Banks can avoid the decline of their profit by making more efficient placements with the available sources. In these conditions, the banking institutions have to apply more effort and be more efficient to make their role as intermediaries sustainable. Banks can respond to the shock caused by negative interest rates by the decomposition of interest and non-interest flows [5] and straightening the main placements witch are generating these flows. These placements are the loans and investments. By increasing the efficiency of these placements, banks can improve the interest and non-interest flows.

The research problem addressed our paper is the determination of the efficiency of creating loans and investment placements of the West Balkan banking institutions in order to ensure the sustainability of net incomes of the banking institutions whose existence has been called into question by the negative interest rates. The second research problem addressed was the determination of the reasons of the various levels of bank efficiencies in the West Balkan region.

The aim of this paper was, in the first stage, to determine the efficiency of banks as intermediaries, in the West Balkan countries, using Data Envelopment Analysis (DEA). After defining the efficiency scores, the aim, in the second stage, was to determine the causes of the different efficiency levels using Tobit's regression model. These two-stage analysis should result in a proposal to improve the input variables in order to achieve higher levels of bank efficiencies to sustain the intermediary function of banks in West Balkan countries. Since there is lack of research papers dealing with the efficiency of banks operating in the Western Balkans, the contribution of this paper is to fill this existing gap. Having in mind the trend of withdrawal of foreign banks in this region and the enlargement of the banking market, it is justified to analyze the efficiency of existing banks in the light of their sustainability on this market.

Measuring bank efficiency using Data Envelopment Analysis (DEA) methodology draws attention in recent years by the academic community. DEA analysis can be used for measuring efficiency in different fields of the financial service industry. The main goal of the analysis is to provide a reliable basis for decision making. Since the focus of the work is the intermediary function of the bank, we used the intermediary approach of the Data Envelopment Analysis (DEA), which is relevant which due to the selection of input and output variables in the first stage.

Our research was conducted in two phases. In the first stage, we analyzed the efficiency of the West Balkan banking institutions in the period from 2015 to 2019 in order to determine in which country the banking institutions are most effective. For the purpose of this analysis, we used the Data Envelopment Analysis. In the second stage of the analysis, we focused on the causes of the different levels of bank efficiencies using Tobit regression analysis.

The contribution of our work is that we used the two-stage model to point out the causes of the various levels of bank efficiencies when it comes to generating loans and investments (intermediary approach). Previous research has focused on the different approaches of the bank efficiencies; on the contrary, we opted for one approach and focused on the analysis of the influencing factors on the efficiency of banks' intermediary function, since it has been questioned by the fall in interest rates on deposits.

Our paper is comprised of five parts. In the Section 1, a brief overview of the research problem and work objectives is provided. The Section 2 provides an overview of the previously published research on which we based our model construction. The Section 3 section is divided into three parts. The first and second parts show the methodology and model used in the first stage and second stage of our research, respectively. The third part provides an overview of the used data. The Section 4 part is separated into two parts. One shows the results of the DEA analysis and the second shows the results of the Tobit's regression model. This part also includes the discussion, which determines the compatibility of our findings with previously conducted research in order to validate our results. The Section 5 part provides concrete guidelines for efficiency improvements and recommendations for further research.

## 2. Theoretical Background

Since the first application of the data envelopment analysis (DEA) by Charnes, Cooper, and Rhodes [6], DEA analyses have been used in all field of economics [7–9]. The primary focus of DEA application is to improve the outputs of the decision making unites (DMU). Three basic DEA models are in use: radial, additive, and slack-based measure models. Recently, DEA models appear more and more for measuring efficiencies of micro-financial institutions like banks, insurance companies, and financial holding companies, etc. [10–14].

As DEA analysis is widely used in different industries, it also has a practical application in the banking service industry. Depending on the requirements of the decision making unit, various modifications of the DEA models can be used. In the finance service industry, DEA is mainly used to measure efficiency on multiple levels in order to provide a reliable basis for decision making depending on the used approach.

In the banking service industry, three major approaches have been singled out [10,15]. The first approach is mainly used for the analysis of branch efficiency [16,17]. This approach is called the service-oriented approach and analyzes the cost efficiency of the bank's branches. This application of the DEA is most suitable for bank managers in cases when a decision is made on the sustainability of the bank's branches. The second approach pays attention to the bank's ability to convert deposits into placements which are generating profits [18]. This is the intermediation approach which analyzes the multiplication function of the bank with respect to the efficiency of the bank to convert deposit sources into placements. The third approach is the revenue or profit-oriented approach, which deals with the efficiency of generating profit in the banking institutions. These three approaches are complementary and can be used jointly [19–21] or separately depending on the research problem. The first approach is used for bank-level analysis and it is best suited for the analysis of an individual bank's branches. On the contrary, the second and the third approaches are used mainly between banks and for cross-country comparisons of banks [13,22]. Some authors add the fourth, operating approach.

Besides the mentioned approaches, in the finance literature, the application of two stage models of DEA is often used [8,23–25]. The term "two-stage model" is not clearly defined in the literature and is used to define various models which include DEA analyses, so it can lead to confusion [26]. Namely, the network DEA models are named two-stage models as well as two-stage analysis, which include DEA efficiency analyses in the first stage and some regression models like Tobit, OLS, and AHP ANN analysis in the second stage [27–31]. The network DEA models are the internal two-stage models and the secondly mentioned two-stage models are the external two-stage models [26].

In the internal two-stage model structure, in the first stage, outputs are generated, using the available inputs, and these outputs of the first stage become the inputs to the second stage [24]. Network DEA models can analyze various efficiencies. For instance, they can measure the profitability efficiency and marketability efficiency [12,32–34]. The first efficiency measure is regarding the ability of generating profit in the first stage. The second efficiency is measured in the second stage and refers to the bank's success in increasing its market value.

In other research papers, network DEA models use cost efficiency and productive efficiency measures [35]. In the first stage, the branch efficiency is measured by using number of branches and number of employees as input variables and using administrative and personnel expenses as output variables. In the second stage of this network DEA model, the output variables of the first stage are used as input and equity and permanent assets are used as outputs to measure the bank's productivity.

Wang et al. [36] used the network DEA approach to measure the efficiencies of 16 major Chinese commercial banks using two-stage analyses. In the first stage they evaluated the deposit-producing process and in the second stage the profit-earning process. In the first stage, the input variables were fixed assets and the number of employees while the output variable was the amount of bank deposits. The amount of bank deposits was the intermediate input/output variable, because it was is in the same time as the input variable of the next stage. The second stage outputs were non-interest incomes, interest incomes, and one undesirable output non-performing loan. The first-stage analysis used the intermediary approach and the second stage the profitability approach.

External two-stage DEA models rely on a combination of DEA methods and further analysis using a regression model. The results of DEA analysis are the dependent variable in the regression analysis.

Barth et al. [37] used an external two-stage model of DEA to measure the branch efficiency of a local German bank. They analyzed the impact of the environmental determinants on branch efficiency, using customer potential and branch characteristic and competitive environment indexes as external variables. The results of their research showed that 8 of 25 branches were wrongly classified as efficient by the traditional model. Regarding the branch efficiency Wu, Yang, and Liang [38] analyzed the branch efficiency of a Canadian bank using the combination of DEA and a Neural Network.

Dar, Ahn, and Dar [29] used DEA analysis and the Tobit regression model in order to measure the central bank efficiency. Their results showed that the export level of the country significantly affects the central bank efficiency.

Paleckova [28] combined DEA and OLS methods in the second stage. In the second stage, internal and external variables were combined in order to determine which one significantly affects the cost efficiency of Slovak and Chech banks. The combination of internal and external variables was also used by Sufian [31]. He used the external two-stage DEA while examining the efficiency of the Malasiyan banking sector. He measured the technical, pure technical, and scale efficiency of individual banks in the first stage then he examined the changes in the efficiency levels before and after merger periods. After that, he applied the multivariate Tobit regression analysis wherein he used internal and external independent variables to measure the impact on the DEA efficiency scores.

External two-stage models are used in cases when the researchers aim to show which determinants have led to certain efficiency scores. In accordance with the mentioned research problem, we used the external two-stage DEA model. In the literature, there are several studies dealing with efficiency in the banking industry, but they are mainly considering the cost efficiency and the profit efficiency, which is, from a practical point of view, crucial. These studies do not have the external two-stage level analysis. There are not enough studies investigating the intermediary approach on the two-stage level. From the theoretical point of view, the intermediary approach is important in sense of the sustainability of the banking institutions among other financial institutions. Considering the lack of literature in this field, the results of our research fill part of that gap.

## 3. Methodology and Data

### 3.1. First-Stage Methodology

Data Envelopment Analysis (DEA) is a non-parametric approach to efficiently analyze and compare the efficiency of each observed decision making unit (DMU) with the highest achieved level of efficiency in the sample. It does not require a priori assumptions about the analytical form of the selected variables. A valuable advantage of the DEA method is that various numbers of heterogeneous input and output variables can be included in the model, which can be represented by different types of metrics [39]. Values of efficiency scores highly depend on selection of sample and variables; therefore, the results of the DEA model show relative efficiency measures [40].

In order to apply DEA correctly, the following conditions must be met:

- The subjects of efficiency assessment are several decision making units (DMUs) with their input and output data [41].
- DMUs, when precisely defined, are always of the same type of institutions (in our paper, e.g., banks).
- DMUs operate within similar business conditions, but often the initial data for different units are significantly different.
- The number of decision-making units should be at least two or three times higher than the sum of diverse inputs and outputs [42], which is confirmed by numerous examples from the literature in which efficiency measurement was performed.
- Flexible DMUs in terms of the suggestions based on DEA results (reduction/increase of a certain input and output) [43].

Recently, numerous variants of the DEA model have been developed. DEA models can be input or output oriented, depending on the demand of the decision making unit. Firstly,

DEA can measure the ability to maximize the outputs without any modification of the inputs. Secondly, it can measure the achieved levels of the output by minimizing the input variables. In this paper, the output-oriented DEA model with a variable return to scale was applied to analyze the efficiency level of banks operating in the Western Balkan countries. The choice of DEA model orientation depends on whether decision-makers have more influence on improving input or output levels; therefore, in our case, the output-oriented model has been chosen. In 1978, Charnes et al. [6], for the first time, introduced a DEA model that allows only a constant return to scale. A few years later, Banker et al. [44] developed a DEA model with a variable return to scale. The application of DEA models with a constant return to scale is appropriate only in cases when all entities operate under the condition of their optimal size [45]. Therefore, in this paper, a DEA model with a variable return to scale was chosen and applied. The analysis was performed by solving the following model (developed by Banker, Charnes, and Cooper in 1984 [44]) of linear programming for each DMU and each period of time:

$$
\begin{aligned}
&\max \phi \\
s.t. \ &\sum_{j=1}^{n} x_{ij}\lambda_j \leq x_{io} \quad i = 1, 2, \ldots, m; \\
&\sum_{j=1}^{n} y_{rj}\lambda_j \geq \phi y_{ro} \quad r = 1, 2, \ldots, s; \\
&\sum_{j=1}^{n} \lambda_j = 1 \\
&\lambda_j \geq 0
\end{aligned}
\tag{1}
$$

where $n$ is the number of entities (DMUs) in the sample (banks in every country in this paper) and $DMU_o$ represents the country under evaluation. $S$ is number of output variables, while $m$ is number of input variables. Observed output and input values are $y_r$ and $x_i$; therefore, $y_{ro}$ is the output $r$ used by $DMU_o$, while $x_{io}$ is the input $i$ used by $DMU_o$. $\lambda$ is the weight of an entity (DMU). Efficiency score is $\phi$.

The focus of our research was the intermediary approach, which refers to the analysis of generating placements (loans and investments) based on available resources of the bank such as labor, capital, and deposits. The input variables in this study were chosen based on previous research which deals with the efficiency of bank placement.

Labor cost or number of employees are the common variables used as input variables in research. Zimkova, Kocisova, Wang, and Barros et al. [15,36,46,47] use the number of full employees as input variables. In contrary to that, Řepková, Svitalkova, Jemric, Chen, and Sufian [18,21,31,48–50] include labor cost in their models. Wanke and Barros [35] included both the number of employees in the first stage as the input variable and personal expenses as the intermediate input/output variable in their internal two-stage model. In that way, both variables were included in the model which deals with the cost efficiency. In our study, which deals with the measuring of the efficiency of creating placements, the number of employees was not relevant, but labor expenses was.

In the analysis of the intermediary function, researchers include either capital or total assets. Since the banks engage capital in order to maximize the returns on it, and on the other side not all assets are generating returns, we are of the same opinion as Jemric and Sufian [21,51] and used capital as a more objective variable.

When it comes to deposits, researchers use them as well as input and output variables [52], depending on the used approach in the research. Since we investigated the intermediary approach, deposits were used as input variables because they generate the pull of funds used for bank placements. In this research, we share the attitude of the researchers who dealt with similar problems [47,53,54].

The output variables were loans and investments, as they are the most common placements of commercial and investment banks which are operating in West Balkan countries. We based our approach to our research problem on the intermediary approach used by Sufian [51].

### 3.2. Second-Stage Methodology

Notwithstanding the frequent remarks/criticism concerning its application [55], the most commonly used model in the second stage of DEA analysis is the censored regression, known as Tobit regression. The second stage of DEA analysis emerges to exemplify the drivers of the technical efficiency results. The results, i.e., values of the efficiency scores, of DEA model lie between 0 and 1. Therefore, the form of the regression model for a limited dependent variable is implemented to determine the relationship between the score and driving aspects of efficiency. Determination of the threshold of the latent dependent variable is important for appropriate censorship of the dependent variable. The common specification of the Tobit model is given as follows [56]:

$$
\begin{aligned}
y_i^* &= x_i'\beta + \varepsilon_i, \\
y_i &= 0 \text{ if } y_i^* \leq 0 \\
y_i &= y_i^* \text{ if } y_i^* \geq 0
\end{aligned}
\tag{2}
$$

where $y_{it}$ is the dependent variable measured by $y_{it}^*$ as the latent dependent variable of the technical efficiency result for positive values and censored otherwise, related to the $i$th country and $t$th year, $x_{it}'$ is the vector of explanatory variables, $\beta$ is a vector of estimable coefficients, and $\varepsilon_{it}$ is a normally and independently distributed error term. Presented model specification is a general dynamic or panel data Tobit model that implements temporal and spatial scale data at the same time. Consequently, two potential model forms are available in terms of omitted effects correlated with the explanatory variables. With the intention of checking the effects, the analysis applies a modified Hausman test with the null hypothesis of using the random effects estimator to run an analysis, while the alternative one recommends to use fixed effect estimator. Alternatively, the null hypothesis implies that there is no correlation between the unique errors and the regressors in the model.

The independent variables in this model are size, merger and acquisition activity, and the type of banks. These variables were chosen to prove whether there are differences in efficiency levels depending on the bank specific determinants. At the same time, the limitation of our model is that it does not consider external determinants influencing the efficiency levels.

There are different stand-out points considering the size of the financial institutions and their efficiencies. On one side, research has shown that small banks have higher efficiency levels [57–59]; on the contrary, some of them show higher efficiencies of the bigger banking institutions [60,61].

In the trends of bank globalization [62], the West Balkan region is characterized by frequent merger and acquisition activities, which certainly affects the efficiency of banks. In our study, we wanted to determine whether mergers and acquisitions have a positive or negative impact on bank efficiency.

It has been shown that different bank types have various levels of efficiencies [54]. Since, in the considered region, there are mostly foreign banks, we examined the types of banks according to the criteria of the bank's specialization,

In accordance with our research problems, we used the intermediary approach of the DEA analysis to examine the efficiency of banks operating in the West Balkan region. In order to determine which variables are affecting the efficiency scores, we applied the Tobit regression analysis in the second stage. Our model was constructed as follows (Figure 1):

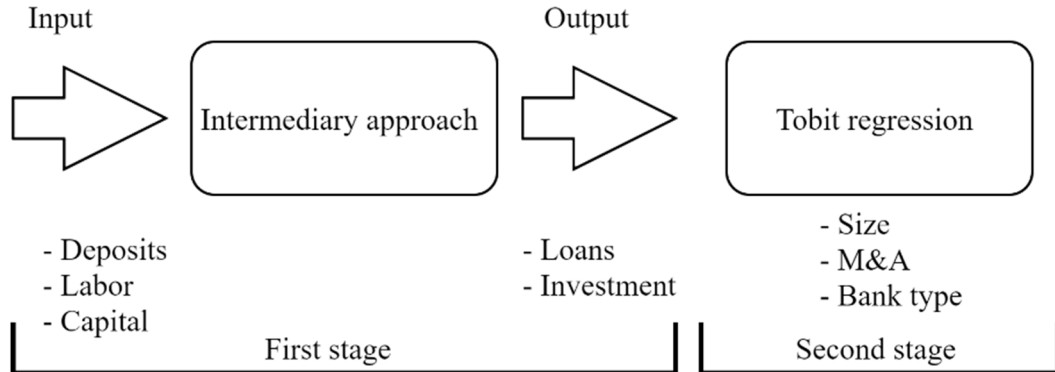

**Figure 1.** Model and research construction. Source: Authors' illustration.

### 3.3. Data and Descriptive Statistics

In order to conduct research related to the achieved level of technical efficiency of banks in Western Balkan countries, the basic characteristics were studied and data were collected from the Annual Reports published at the websites of The National Banks for the selected five countries. The analysis was performed separately for every year and every country in the observed time period of five years, from 2015 until 2019. This research includes the analysis of 78 banks in the West Balkan region (number of banks by country is given in Appendix A). In the abovementioned output-oriented DEA model with variable return to scale, three input and two output variables were selected (Table 1). Deposits, labor costs, and capital were used as input variables, while loans and investments were output variables. While we used the intermediary approach, deposits were used as input variables [52]. All variables were presented in national currencies. Descriptive statistics for selected inputs and outputs are presented in Table 1 for the first and the last year in the sample.

**Table 1.** Descriptive statistics of input and output variables.

| Country | Year | | Deposits | Labor Costs | Capital | Loans | Investment |
|---|---|---|---|---|---|---|---|
| Albania | 2019 | Average | $1.08 \times 10^8$ | 1,260,177 | 14,052,888.83 | 69,770,631 | 27,533,640 |
| | | St. dev. | 98,441,986 | 1,367,652 | 12,227,157.55 | 68,412,564 | 24,782,725 |
| | 2015 | Average | 90,092,952 | 890,502.4 | 34,625,635.50 | 54,687,811 | 20,539,133 |
| | | St. dev. | 95,523,979 | 748,557.5 | 84,589,667.66 | 60,200,729 | 19,404,261 |
| Bosnia and Herzegovina | 2019 | Average | 1,264,682 | 16,146.25 | 206,289.06 | 931,199.1 | 82,138.75 |
| | | St. dev. | 1,476,473 | 14,725.44 | 217,606.27 | 991,735.6 | 138,146.8 |
| | 2015 | Average | 844,810.6 | 14,823.13 | 167,476.38 | 670,763.8 | 43,664.31 |
| | | St. dev. | 1,004,589 | 14,321.3 | 192,804.82 | 757,179 | 104,953 |
| North Macedonia | 2019 | Average | 31,791,816 | 373,602.6 | 5,325,674.33 | 24,664,780 | 4,724,654 |
| | | St. dev. | 36,356,007 | 345,829.9 | 5,871,199.43 | 24,311,119 | 6,290,253 |
| | 2015 | Average | 22,553,986 | 318,252.5 | 3,798,362.62 | 19,155,675 | 3,125,367 |
| | | St.dev. | 27,985,116 | 278,834.2 | 4,586,664.36 | 20,811,576 | 4,252,715 |
| Montenegro | 2019 | Average | 333,511.9 | 5891.5 | 57,535.92 | 264,095.5 | 43,547.08 |
| | | St.dev. | 229,104.9 | 5145.2 | 58,546.66 | 219,775.4 | 44,111.54 |
| | 2015 | Average | 253,910.1 | 5159.09 | 47,611.64 | 207,445.7 | 22,525.91 |
| | | St.dev. | 198,184.7 | 3544.74 | 45,594.37 | 162,602.5 | 27,174.42 |
| Serbia | 2019 | Average | $1.25 \times 10^8$ | 1,722,804 | 27,143,917.38 | $1 \times 10^8$ | $1.03 \times 10^8$ |
| | | St.dev. | $1.35 \times 10^8$ | 1,578,428 | 28,744,289.01 | $1.04 \times 10^8$ | $1.1 \times 10^8$ |
| | 2015 | Average | 84,285,343 | 1,445,207 | 22,853,852.48 | 46,058,771 | 22,386,434 |
| | | St.dev. | $1 \times 10^8$ | 1,357,364 | 28,173,476.05 | 57,787,377 | 33,011,840 |

Source: Authors' calculation.

## 4. Results and Discussion

### 4.1. DEA Efficiency Results

Data analysis was performed using DeaMax software using the variable return-to-scale DEA method with output orientation. This means that we wanted to see which

banks operate efficiently and which banks need to increase their efficiency by achieving a higher level of output variables. The results obtained from the output-oriented DEA model with a variable return to scale are shown in Figure 2 and Appendix A Table A1. From the presented results, it can be concluded that banks in the Western Balkan countries operate at an enviable level of efficiency since the average score was above 85% in the observed time period.

Montenegro consistently had the highest efficiency score during those five years, followed by Bosnia and Herzegovina with values higher than 97%. Albania was the only country that had an average efficiency score below 85% in 2015, but with an increase in 2016 and 2018 achieved the maximum average efficiency score in 2019. Banks in North Macedonia also showed an increasing trend in average efficiency score, while the average efficiency score in Serbia had a decline below 90% in 2016. In the last observed year, the average efficiency score for all Western Balkan countries was above 95%.

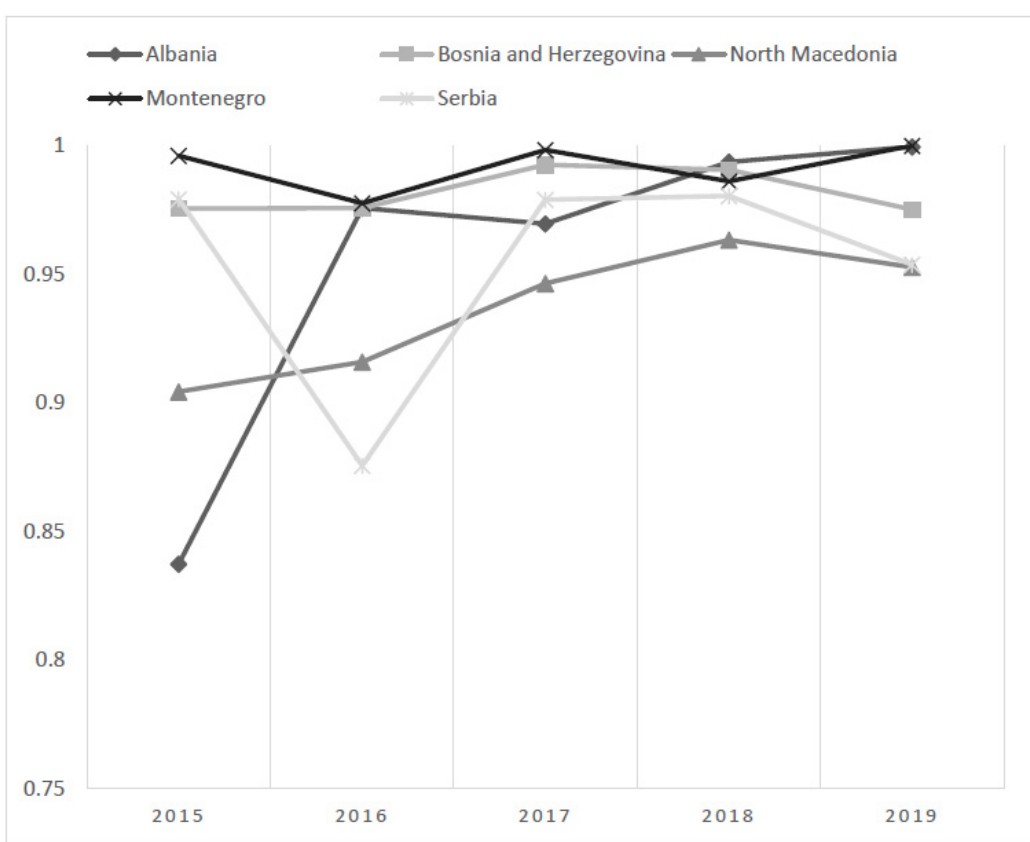

**Figure 2.** Average efficiency score per year per country. Source: Authors.

*4.2. Tobit Regression Results and Disscusion*

Furthermore, the Tobit model analyzes the size and direction of the relative effect of the external factors (explanatory variables) in terms of their impact on the relative efficiency scores (dependent variable). External variables are not decision variables that have already been included in the DEA analysis. In particular, three dummy variables were used in this paper as a group of explanatory variables.

$$\theta_{it} = \alpha + \beta_1 \cdot CI_{it} + \beta_2 \cdot MA_{it} + \beta_3 \cdot SIZE_{it} + \varepsilon_{it} \tag{3}$$

where,

　　$\theta$ represents the relative technical efficiency score (dependent variable),
　　$i$ is a country,
　　$t$ is the time period (in years),
　　$\alpha$ is the model constant coefficient,

*βs* are the model coefficients for the explanatory variables.

The explanatory variables in the model (3) are set as follows:

- CI—dummy variable for commercial or investment banks (0—commercial banks, 1—investment bank)
- MA—dummy variable for merger and acquisitions (0—periods without merger and acquisitions, 1—periods with merger and acquisitions)
- SIZE—dummy variable for division into a group of small and large banks (0—small bank, 1—large bank)

The estimation results of the Tobit regression model applying a random-effects form of the model are illustrated in Table 2. Additionally, a modified Hausmann test result $\chi 2 = 1.4183$ specifies the application of a random effects form of the model instead of a fixed-effects form. Additionally, with the acceptance of the null hypothesis, the test result showed that there is no endogeneity problem in the model.

**Table 2.** Coefficients and test values of the Tobit model.

| Variable | Coefficient | z-Statistic |
|---|---|---|
| Constant | 0.95968 *** | 140.689 |
| Commercial or investment bank | −0.00319 * | −1.74597 |
| Merger and acquisition | −0.01525 ** | −2.39716 |
| Bank size | 0.03267 *** | 2.77225 |

Note: ***, **, and * indicate the significance at the level of 1%, 5%, and 10%. $\chi^2 = 1.4183$.

The estimated parameters of Tobit model, presented in Table 2, indicate the existence of significance of the dummy variable for commercial or investment banks, dummy variable for merger and acquisitions, and dummy variable for division into a group of small and large banks. The dummy variable for commercial or investment banks had a significant and negative influence on relative technical efficiency of banks in the observed period of time, suggesting a higher level of relative efficiency in the case of commercial banks. Such results were expected since mostly commercial banks generate placements based on deposits. Investment banks have non-deposit sources of financing and are not focused on collecting deposits [63]. The coefficient for the dummy variable for merger and acquisitions remained significant and negative with overall efficiency results, showing that banks achieved lower relative efficiency performance at the time of the takeover. These banks, operating in West Balkan countries, were the targets of the takeover and it is important to notice that in the next period of time an improvement in their efficiency is to be expected as a result of the synergetic effect [64]. Sufian [31] also suggested in his research that M&A activity resulted in the improvement in the period after the takeover. Authors [65–67] proved that inefficient banks are more likely targets of cross-border merger and acquisitions. Shi et al. [44] suggested that full-scale merger of bank is not proper. The decomposition of the merger efficiency into technical, harmony, and scale efficiency allows the identification of alternative ways to improve merger performances in deposit and profit-earning process. Furthermore, another important determinant of relative bank efficiency is a dummy variable for bank size. Coefficient indicates a positive and significant relationship between bank size and relative efficiency at the significance level of 1%. There are a lot of debatable results regarding the relationship between bank size and its relative efficiency. Delis and Papanikolaou [60] showed that bank size has a positive significant economic and statistical effect on bank efficiency only when semi-parametric models are engaged. Kumar and Singh [45] believe that large deposits and operating costs can lead to lower relative efficiencies. Moutinho et al. [46] presented inconclusive results for the size variable, showing a negative and significant effect of that size variable on banks' efficiency in some of the estimated models, and also a positive and significant effect of size on banks' efficiency, considering other estimated models. Ouenniche and Carrales [47] suggested that efficiency profiles are quite different when it comes to the size of bank. In fact, authors noticed that large banks are more technically efficient overall than the small ones [68,69],

while the large ones seem to be less scale efficient than small banks [70]. The results of our study showed that lager banks are more efficient in generating loans and investments. These results are expected from the theoretical point of view, because larger bank have a higher deposit potential which generates these placements. In previously conducted studies, the cost and profit earning efficiencies were in focus.

## 5. Conclusions

The results of our research showed that two of five considered countries had less fluctuations in the observed time period. Namely, Serbia, Northern Macedonia, and Albania had difficulties with the efficiency of generating loans and investment in 2016, 2015–2016, and 2015, respectively. The reason for that is a smaller market share of the banks in the initial years of the observed period. The Western Balkans financial market has been quite turbulent in the last decade in terms of mergers and acquisitions. Several banking groups withdrew from this market and other banking groups took over their market share. This is one of the reasons for the increased efficiency levels in the considered region.

Our research showed that commercial banks are more efficient in the intermediary function than investment banks. Regarding the size, our finding proved that larger banks are more successful in intermediation than small banks. This finding substantiates the fact that the banking system showed better efficiency scores in the last years of the observed period. Merger and acquisition activities in the region support the claim that inefficient banks have been taken over by larger banks that have proven to be more efficient when it comes to intermediation. This brings us to the conclusion that large commercial banks can sustain on the financial market of the West Balkan region. Small banks, which were proven to be less efficient, are highly likely to become the target of large banks that will take them over. Therefore, in the future, a reduction in the number of banks that will have a higher market share can be expected in this market. In such a market environment, it is interesting to examine the relationship between banks and other financial institutions and to determine the efficiency of banks in relation to other financial institutions. Further research could be directed on extending this research to the financial market.

**Author Contributions:** This research article is the result of common work of the listed authors. Each author has his own contribution to the work as follows: Conceptualization, N.M. and B.R.; methodology, B.R. and A.M.H.; software, B.R. and A.M.H.; validation, B.R.; formal analysis, N.M., B.R., B.K. and A.M.H.; resources, N.M., B.R. and B.K.; data curation, B.K. and N.M.; writing—original draft preparation, N.M., B.R., B.K. and A.M.H.; writing—review and editing, N.M.; visualization, N.M.; supervision, N.M., B.R., B.K. and A.M.H. All authors have read and agreed to the published version of the manuscript.

**Funding:** This research received no external funding.

**Institutional Review Board Statement:** Not applicable.

**Informed Consent Statement:** Not applicable.

**Data Availability Statement:** Data used in this research were retrieved from the revised financial statements of the banks included in the sample. The financial statements are available on the official websites of the banks operating in the West Balkan countries (Serbia, Bosnia and Herzegovina, North Macedonia, Montenegro, and Albania). The list of active banks operating in the mentioned region is available on the websites of the national banks of the analyzed countries as of May 2021.

**Conflicts of Interest:** The authors declare no conflict of interest.

## Appendix A

Appendix A presents efficiency scores of DEA model. The efficiency scores are shown for all banks in all considered countries for the period from 2015 to 2019. The summary of Appendix A is shown on the Figure 2.

**Table A1.** DEA efficiency model scores.

| 2015 | | 2016 | | 2017 | | 2018 | | 2019 | |
|---|---|---|---|---|---|---|---|---|---|
| **DMU** | **Score** | **DMU** | **Score** | **DMU** | **Score** | **DMU** | **Score** | **DMU** | **Score** |
| Albania1 | 0.958 | Albania1 | 0.991 | Albania1 | 0.921 | Albania1 | 1.000 | Albania1 | 1.000 |
| Albania2 | 0.557 | Albania2 | 1.000 | Albania2 | 1.000 | Albania2 | 1.000 | Albania2 | 1.000 |
| Albania3 | 1.000 | Albania3 | 1.000 | Albania3 | 1.000 | Albania3 | 1.000 | Albania3 | 1.000 |
| Albania4 | 1.000 | Albania4 | 1.000 | Albania4 | 1.000 | Albania4 | 1.000 | Albania4 | 1.000 |
| Albania5 | 0.712 | Albania5 | 1.000 | Albania5 | 1.000 | Albania5 | 1.000 | Albania5 | 1.000 |
| Albania6 | 1.000 | Albania6 | 1.000 | Albania6 | 1.000 | Albania6 | 1.000 | Albania6 | 1.000 |
| Albania7 | 0.837 | Albania7 | 0.979 | Albania7 | 1.000 | Albania7 | 1.000 | Albania7 | 1.000 |
| Albania8 | 1.000 | Albania8 | 1.000 | Albania8 | 1.000 | Albania8 | 1.000 | Albania8 | 1.000 |
| Albania9 | 0.564 | Albania9 | 1.000 | Albania9 | 1.000 | Albania9 | 1.000 | Albania9 | 1.000 |
| Albania10 | 0.805 | Albania10 | 1.000 | Albania10 | 1.000 | Albania10 | 0.923 | Albania10 | 0.995 |
| Albania11 | 0.615 | Albania11 | 0.737 | Albania11 | 0.714 | Albania11 | 1.000 | Albania11 | 1.000 |
| Albania12 | 1.000 | Albania12 | 1.000 | Albania12 | 1.000 | Albania12 | 1.000 | Albania12 | 1.000 |
| Average | 0.837 | | 0.976 | | 0.970 | | 0.994 | | 1.000 |
| Bosnia&Herzegovina1 | 0.608 | Bosnia&Herzegovina1 | 0.630 | Bosnia&Herzegovina1 | 1.000 | Bosnia&Herzegovina1 | 1.000 | Bosnia&Herzegovina1 | 1.000 |
| Bosnia&Herzegovina2 | 1.000 | Bosnia&Herzegovina2 | 1.000 | Bosnia&Herzegovina2 | 0.926 | Bosnia&Herzegovina2 | 0.998 | Bosnia&Herzegovina2 | 0.932 |
| Bosnia&Herzegovina3 | 1.000 | Bosnia&Herzegovina3 | 0.981 | Bosnia&Herzegovina3 | 0.954 | Bosnia&Herzegovina3 | 1.000 | Bosnia&Herzegovina3 | 1.000 |
| Bosnia&Herzegovina4 | 1.000 | Bosnia&Herzegovina4 | 1.000 | Bosnia&Herzegovina4 | 1.000 | Bosnia&Herzegovina4 | 1.000 | Bosnia&Herzegovina4 | 1.000 |
| Bosnia&Herzegovina5 | 1.000 | Bosnia&Herzegovina5 | 1.000 | Bosnia&Herzegovina5 | 1.000 | Bosnia&Herzegovina5 | 1.000 | Bosnia&Herzegovina5 | 1.000 |
| Bosnia&Herzegovina6 | 1.000 | Bosnia&Herzegovina6 | 1.000 | Bosnia&Herzegovina6 | 1.000 | Bosnia&Herzegovina6 | 1.000 | Bosnia&Herzegovina6 | 1.000 |
| Bosnia&Herzegovina7 | 1.000 | Bosnia&Herzegovina7 | 1.000 | Bosnia&Herzegovina7 | 1.000 | Bosnia&Herzegovina7 | 0.850 | Bosnia&Herzegovina7 | 0.732 |
| Bosnia&Herzegovina8 | 1.000 | Bosnia&Herzegovina8 | 1.000 | Bosnia&Herzegovina8 | 1.000 | Bosnia&Herzegovina8 | 1.000 | Bosnia&Herzegovina8 | 1.000 |
| Bosnia&Herzegovina9 | 1.000 | Bosnia&Herzegovina9 | 1.000 | Bosnia&Herzegovina9 | 1.000 | Bosnia&Herzegovina9 | 1.000 | Bosnia&Herzegovina9 | 1.000 |
| Bosnia&Herzegovina10 | 1.000 | Bosnia&Herzegovina10 | 1.000 | Bosnia&Herzegovina10 | 1.000 | Bosnia&Herzegovina10 | 1.000 | Bosnia&Herzegovina10 | 1.000 |
| Bosnia&Herzegovina11 | 1.000 | Bosnia&Herzegovina11 | 1.000 | Bosnia&Herzegovina11 | 1.000 | Bosnia&Herzegovina11 | 1.000 | Bosnia&Herzegovina11 | 0.947 |
| Bosnia&Herzegovina12 | 1.000 | Bosnia&Herzegovina12 | 1.000 | Bosnia&Herzegovina12 | 1.000 | Bosnia&Herzegovina12 | 1.000 | Bosnia&Herzegovina12 | 1.000 |
| Bosnia&Herzegovina13 | 1.000 | Bosnia&Herzegovina13 | 1.000 | Bosnia&Herzegovina13 | 1.000 | Bosnia&Herzegovina13 | 1.000 | Bosnia&Herzegovina13 | 1.000 |
| Bosnia&Herzegovina14 | 1.000 | Bosnia&Herzegovina14 | 1.000 | Bosnia&Herzegovina14 | 1.000 | Bosnia&Herzegovina14 | 1.000 | Bosnia&Herzegovina14 | 1.000 |
| Bosnia&Herzegovina15 | 1.000 | Bosnia&Herzegovina15 | 1.000 | Bosnia&Herzegovina15 | 1.000 | Bosnia&Herzegovina15 | 1.000 | Bosnia&Herzegovina15 | 1.000 |

**Table A1.** *Cont.*

| 2015 | | 2016 | | 2017 | | 2018 | | 2019 | |
|---|---|---|---|---|---|---|---|---|---|
| **DMU** | **Score** | **DMU** | **Score** | **DMU** | **Score** | **DMU** | **Score** | **DMU** | **Score** |
| Bosnia&Herzegovina16 | 1.000 | Bosnia&Herzegovina16 | 1.000 | Bosnia&Herzegovina16 | 1.000 | Bosnia&Herzegovina16 | 1.000 | Bosnia&Herzegovina16 | 0.992 |
| Average | 0.976 | | 0.976 | | 0.993 | | 0.990 | | 0.975 |
| North Macedonia1 | 1.000 | North Macedonia1 | 1.000 | North Macedonia1 | 1.000 | North Macedonia1 | 1.000 | North Macedonia1 | 1.000 |
| North Macedonia2 | 0.795 | North Macedonia2 | 0.996 | North Macedonia2 | 0.977 | North Macedonia2 | 1.000 | North Macedonia2 | 1.000 |
| North Macedonia3 | 0.987 | North Macedonia3 | 0.764 | North Macedonia3 | 0.922 | North Macedonia3 | 1.000 | North Macedonia3 | 0.913 |
| North Macedonia4 | 1.000 | North Macedonia4 | 1.000 | North Macedonia4 | 1.000 | North Macedonia4 | 1.000 | North Macedonia4 | 1.000 |
| North Macedonia5 | 1.000 | North Macedonia5 | 1.000 | North Macedonia5 | 1.000 | North Macedonia5 | 0.999 | North Macedonia5 | 0.956 |
| North Macedonia6 | 1.000 | North Macedonia6 | 1.000 | North Macedonia6 | 1.000 | North Macedonia6 | 1.000 | North Macedonia6 | 1.000 |
| North Macedonia7 | 1.000 | North Macedonia7 | 1.000 | North Macedonia7 | 1.000 | North Macedonia7 | 1.000 | North Macedonia7 | 1.000 |
| North Macedonia8 | 1.000 | North Macedonia8 | 1.000 | North Macedonia8 | 1.000 | North Macedonia8 | 1.000 | North Macedonia8 | 1.000 |
| North Macedonia9 | 0.506 | North Macedonia9 | 0.630 | North Macedonia9 | 0.784 | North Macedonia9 | 0.839 | North Macedonia9 | 0.889 |
| North Macedonia10 | 0.662 | North Macedonia10 | 0.675 | North Macedonia10 | 0.750 | North Macedonia10 | 0.820 | North Macedonia10 | 0.675 |
| North Macedonia11 | 1.000 | North Macedonia11 | 1.000 | North Macedonia11 | 1.000 | North Macedonia11 | 1.000 | North Macedonia11 | 1.000 |
| North Macedonia12 | 0.806 | North Macedonia12 | 0.843 | North Macedonia12 | 0.870 | North Macedonia12 | 0.865 | North Macedonia13 | 1.000 |
| North Macedonia13 | 1.000 | North Macedonia13 | 1.000 | North Macedonia13 | 1.000 | North Macedonia13 | 1.000 | | |
| Average | 0.904 | | 0.916 | | 0.946 | | 0.963 | | 0.953 |
| Montenegro1 | 1.000 | Montenegro1 | 1.000 | Montenegro1 | 1.000 | Montenegro1 | 1.000 | Montenegro1 | 1.000 |
| Montenegro2 | 1.000 | Montenegro2 | 1.000 | Montenegro2 | 1.000 | Montenegro2 | 1.000 | Montenegro2 | 1.000 |
| Montenegro3 | 0.957 | Montenegro3 | 0.841 | Montenegro3 | 0.980 | Montenegro3 | 1.000 | Montenegro3 | 1.000 |
| Montenegro4 | 1.000 | Montenegro4 | 0.950 | Montenegro4 | 1.000 | Montenegro4 | 1.000 | Montenegro4 | 1.000 |
| Montenegro5 | 1.000 | Montenegro5 | 1.000 | Montenegro5 | 1.000 | Montenegro5 | 0.998 | Montenegro5 | 1.000 |
| Montenegro6 | 1.000 | Montenegro6 | 0.940 | Montenegro6 | 1.000 | Montenegro6 | 0.996 | Montenegro6 | 1.000 |
| Montenegro7 | 1.000 | Montenegro7 | 1.000 | Montenegro7 | 1.000 | Montenegro7 | 1.000 | Montenegro7 | 1.000 |
| Montenegro8 | 1.000 | Montenegro8 | 1.000 | Montenegro8 | 1.000 | Montenegro8 | 1.000 | Montenegro8 | 1.000 |
| Montenegro9 | 1.000 | Montenegro9 | 1.000 | Montenegro9 | 1.000 | Montenegro9 | 1.000 | Montenegro9 | 1.000 |
| Montenegro10 | 1.000 | Montenegro10 | 1.000 | Montenegro10 | 1.000 | Montenegro10 | 0.838 | Montenegro10 | 1.000 |
| Montenegro11 | 1.000 | Montenegro11 | 1.000 | Montenegro11 | 1.000 | Montenegro11 | 1.000 | Montenegro11 | 1.000 |
| | | Montenegro12 | 1.000 | Montenegro12 | 1.000 | Montenegro12 | 1.000 | Montenegro12 | 1.000 |

**Table A1.** *Cont.*

| 2015 | | 2016 | | 2017 | | 2018 | | 2019 | |
|---|---|---|---|---|---|---|---|---|---|
| **DMU** | **Score** | **DMU** | **Score** | **DMU** | **Score** | **DMU** | **Score** | **DMU** | **Score** |
| Average | 0.996 | | 0.978 | | 0.998 | | 0.986 | | 1.000 |
| Serbia1 | 1.000 | Serbia1 | 0.920 | Serbia1 | 1.000 | Serbia1 | 1.000 | Serbia1 | 1.000 |
| Serbia2 | 1.000 | Serbia2 | 0.802 | Serbia2 | 1.000 | Serbia2 | 1.000 | Serbia2 | 0.972 |
| Serbia3 | 1.000 | Serbia3 | 1.000 | Serbia3 | 1.000 | Serbia3 | 1.000 | Serbia3 | 0.967 |
| Serbia4 | 1.000 | Serbia4 | 1.000 | Serbia4 | 1.000 | Serbia4 | 1.000 | Serbia4 | 1.000 |
| Serbia6 | 1.000 | Serbia5 | 0.000 | Serbia5 | 1.000 | Serbia5 | 1.000 | Serbia5 | 1.000 |
| Serbia7 | 0.941 | Serbia6 | 0.809 | Serbia6 | 1.000 | Serbia6 | 1.000 | Serbia6 | 0.840 |
| Serbia8 | 1.000 | Serbia7 | 0.961 | Serbia7 | 1.000 | Serbia7 | 1.000 | Serbia7 | 1.000 |
| Serbia9 | 1.000 | Serbia8 | 1.000 | Serbia8 | 1.000 | Serbia8 | 0.911 | Serbia8 | 0.912 |
| Serbia10 | 1.000 | Serbia9 | 1.000 | Serbia9 | 1.000 | Serbia9 | 1.000 | Serbia9 | 1.000 |
| Serbia11 | 0.759 | Serbia10 | 1.000 | Serbia10 | 1.000 | Serbia10 | 1.000 | Serbia10 | 1.000 |
| Serbia12 | 0.939 | Serbia11 | 0.807 | Serbia11 | 0.939 | Serbia11 | 0.978 | Serbia11 | 0.845 |
| Serbia13 | 0.985 | Serbia12 | 0.766 | Serbia12 | 0.951 | Serbia12 | 0.919 | Serbia12 | 0.887 |
| Serbia14 | 1.000 | Serbia13 | 1.000 | Serbia13 | 1.000 | Serbia13 | 0.981 | Serbia13 | 0.865 |
| Serbia15 | 1.000 | Serbia14 | 1.000 | Serbia14 | 1.000 | Serbia14 | 1.000 | Serbia14 | 0.987 |
| Serbia16 | 1.000 | Serbia15 | 0.973 | Serbia15 | 0.900 | Serbia15 | 1.000 | Serbia15 | 1.000 |
| Serbia17 | 0.989 | Serbia16 | 1.000 | Serbia16 | 1.000 | Serbia16 | 1.000 | Serbia16 | 0.931 |
| Serbia18 | 1.000 | Serbia17 | 0.796 | Serbia17 | 0.923 | Serbia17 | 0.926 | Serbia17 | 1.000 |
| Serbia19 | 0.979 | Serbia18 | 1.000 | Serbia18 | 1.000 | Serbia18 | 1.000 | Serbia18 | 1.000 |
| Serbia20 | 0.997 | Serbia19 | 0.662 | Serbia19 | 1.000 | Serbia19 | 1.000 | Serbia19 | 1.000 |
| Serbia21 | 0.967 | Serbia20 | 0.911 | Serbia20 | 1.000 | Serbia20 | 1.000 | Serbia20 | 1.000 |
| Serbia22 | 0.957 | Serbia21 | 0.949 | Serbia21 | 0.820 | Serbia21 | 0.837 | Serbia21 | 0.866 |
| Serbia23 | 1.000 | Serbia22 | 0.829 | Serbia22 | 0.921 | Serbia22 | 0.940 | Serbia22 | 0.975 |
| Serbia24 | 1.000 | Serbia23 | 1.000 | Serbia23 | 1.000 | Serbia23 | 1.000 | Serbia23 | 1.000 |
| Serbia25 | 1.000 | Serbia24 | 0.579 | Serbia24 | 1.000 | Serbia24 | 1.000 | Serbia24 | 0.874 |
| Serbia26 | 0.967 | Serbia25 | 1.000 | Serbia25 | 1.000 | Serbia25 | 1.000 | Serbia25 | 1.000 |
| | | Serbia26 | 1.000 | Serbia26 | 1.000 | Serbia26 | 1.000 | Serbia26 | 0.879 |
| Average | 0.979 | | 0.876 | | 0.979 | | 0.980 | | 0.954 |

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
