# Peer review of "External Two Stage DEA Analysis of Bank Efficiency in West Balkan Countries"

_sustainability, doi:10.3390/su14020978_

Round 1

Reviewer 1 Report

- Many studies explore DEA in the banking industry. The paper should highlight its contributions and differentials to theory and practice, vis-à-vis the other studies.

- In this context, a more thorough literature review is necessary to support the gap studied in the paper.

- In addition, it is important to discuss the variables that were used in the DEA model. There seem to be just a few input and output variables.

- In this context, the database should be better presented, indicating its characteristics and whether other variables were available.

- Other information, i.e., number of banks of each country, differences in regulations among countries, should also be presented. 

- The method should be substantially enhanced. For instance, there are many different aspects of DEA for banks, including scale, orientation, variables related to inputs and outputs, etc. 

- The choices of the DEA model should be discussed and accompanied by supporting references.

- One of the important discussions of DEA in banking is whether some variables, for instance, risk, leverage, should be input or output.

- Therefore, the choice of the inputs and outputs could be supported by references and better discussed. 

- The paper briefly analyzes the potential issues of omitted variables in the second stage. 

- A more insightful discussion should be introduced to deal with problems with endogeneity since a regression analysis is conducted. 

- A more detailed discussion on why those explanatory variables, from a theoretical point of view, is paramount.

Robustness checks should be conducted using different variables (for instance, relative values, instead of absolute values) or models.

- For instance, logistic or probit models yield the same results as Tobit?

Minor issues:

- Table 1 should be reorganized to summarize results better. For instance, it should be better to avoid repetition of wordx (average, standard deviation, name of countries)

- I did not see the number of banks in the study. 

- Are the monetary figures in USD?

Author Response

Thank you for the valuable comments and the effort to help us improve our paper. Our response to the reviewer is given in the table below. 

Reviewer 2 Report

The paper examines bank efficiency in West Balkan countries. However, my comments and suggestions are as follows:

1.The implications should be added at the end of the abstract.

2.In the abstract, the research background is not clear, and the authors should cite some literature to support their opinions.

3.The contributions of this paper should be specific.

4.The discussion should be strengthened.

5.The limitations of this paper are missing.

6.There are some mistakes in this paper. For example,

In the abstract, please provide the full names of “DEA” and “M&A”.

The abbreviation “DEA” should appear in line 42.

Author Response

(The authors gave the same response as above.)

Round 2

Reviewer 2 Report

Thanks for the effors of the authors. I think that the paper can be accepted.